# Enhancement of Multi-Scale Self-Organization Processes during Inconel DA 718 Machining through the Optimization of TiAlCrSiN/TiAlCrN Bi-Nano-Multilayer Coating Characteristics

**DOI:** 10.3390/ma15041329

**Published:** 2022-02-11

**Authors:** Guerman Fox-Rabinovitch, Goulnara Dosbaeva, Anatoly Kovalev, Iosif Gershman, Kenji Yamamoto, Edinei Locks, Jose Paiva, Egor Konovalov, Stephen Veldhuis

**Affiliations:** 1Department of Mechanical Engineering, McMaster Manufacturing Research Institute (MMRI), McMaster University, Hamilton, ON L8S 4L8, Canada; dosby@mcmaster.ca (G.D.); lockse@mcmaster.ca (E.L.); paivajj@mcmaster.ca (J.P.); veldhu@mcmaster.ca (S.V.); 2I.P. Bardin Central Scientific Research Institute for Ferrous Metallurgy (CNIICHERMET), Physical Metallurgy Center, Radio Street 23/9, 105005 Moscow, Russia; a_kovalev@sprg.ru (A.K.); konovalov@sprg.ru (E.K.); 3Joint Stock Company Railway Research Institute, Moscow State Technological University “Stankin” (MSTU “STANKIN”), 127994 Moscow, Russia; isgershman@gmail.com; 4Applied Physics Research Laboratory, Kobe Steel Ltd., 1-5-5 Takatsuda-dai, Nishi-ku, Kobe 651-2271, Hyogo, Japan; yamamoto.kenji1@kobelco.com

**Keywords:** self-organization, tribo-films, nano-multilayer PVD coatings, cutting tools

## Abstract

Optimization of the composition of a new generation of bi-nano-multilayered TiAlCrSiN/TiAlCrN-based coatings is outlined in this study for the machining of direct aged (DA) Inconel 718 alloy. Three types of TiAlCrSiN/TiAlCrN-based bi-nano-multi-layer coatings with varying chemical compositions were investigated: (1) a previous state-of-the-art Ti_0.2_Al_0.55_Cr_0.2_Si_0.03_Y_0.02_N/Ti_0.25_Al_0.65_Cr_0.1_N (coating A); (2) Ti_0.2_Al_0.52_Cr_0.2_Si_0.08_N/Ti_0.25_Al_0.65_Cr_0.1_N with increased amount of Si (up to 8 at.%; coating B); (3) a new Ti_0.18_Al_0.55_Cr_0.17_Si_0.05_Y_0.05_N/Ti_0.25_Al_0.65_Cr_0.1_N coating (coating C) with an increased amount of both Si and Y (up to 5 at.% each). The structure of each coating was evaluated by XRD analysis. Micro-mechanical characteristics were investigated using a MicroMaterials NanoTest system and an Anton Paar-RST3 tester. The wear performance of nano-multilayered TiAlCrSiN/TiAlCrN-based coatings was evaluated during the finish turning of direct aged (DA) Inconel 718 alloy. The wear patterns were assessed using optical microscopy imaging. The tribological performance was evaluated through (a) a detailed chip characteristic study and (b) XPS studies of the worn surface of the coated cutting tool. The difference in tribological performance was found to correspond with the type and amount of tribo-films formed on the friction surface under operation. Simultaneous formation of various thermal barrier tribo-films, such as sapphire, mullite, and garnet, was observed. The overall amount of beneficial tribo-films was found to be greater in the new Ti_0.18_Al_0.55_Cr_0.17_Si _0.05_Y_0.05_N/Ti_0.25_Al_0.65_Cr_0.1_N nano-bi-multilayer coating (coating C) than in the previous state-of-the-art coatings (A and B). This resulted in over two-fold improvement of this coating’s tool life compared with those of the commercial benchmark AlTiN coating and coating B, as well as a 40% improvement of the tool life of the previous state-of-the-art coating A. Multi-scale self-organization processes were observed: nano-scale tribo-film formation on the cutting tool surface combined with micro-scale generation of strain-induced martensite zones as a result of intensive metal flow during chip formation. Both of these processes are strongly enhanced in the newly developed coating C.

## 1. Introduction

Nickel-based superalloys are widely used in aerospace applications due to their unique combination of high strength [1] and chemical/thermal stability [2] at elevated temperatures [3]. However, these beneficial service properties severely reduce the machinability of such alloys. The chief problems encountered during the machining of these alloys are intensive heat generation, low thermal conductivity [3], and heavy loading within the cutting zone. The austenitic matrix of Ni-based superalloys facilitates their reaction with tool materials under atmospheric conditions, thereby accelerating work hardening [1]. As such, these materials have a considerable tendency to adhere and even weld to the cutting tool’s surface [2], resulting in attrition wear [4] and the formation of a non-stable built-up edge (BUE) [5], which eventually lead to severe surface damage [6]. Superalloys also contain abrasive carbide particles in their structure, which contribute to cutting tool flank wear [7]. Therefore, the machining of Ni-based superalloys presents a significant challenge. The introduction of superalloys such as direct aged Inconel 718 with improved temperature strength and reduced thermal conductivity further complicates the machining process [7]. Since less heat is removed from the tool/chip interface under operation, the temperature range at the tool surface can be as high as 900–1000 °C [1,2,6]. Tool materials with high hardness such as carbides, ceramics, and cubic boron nitride are regularly used to machine nickel-based superalloys [8]. Surface engineering of cemented carbide cutting tools can be an effective solution to the problems posed by the machining of these hard-to-cut materials, particularly through the application of Physical Vapor Deposited (PVD) coatings. Various TiAlN- and AlCrN-based hard PVD coatings are widely used for this purpose [9,10]. There exist two design approaches for enhancing the cutting performance of PVD coatings. The first one seeks to improve coating wear performance in terms of its micro-mechanical characteristics and tends to predominate in the scientific literature [11,12,13,14,15,16,17,18]. Nanocomposite coatings are designed according to this approach [19]. They are capable of sustaining heavy loads under operation due to their significantly increased hardness (around 40 GPa or above) even at high temperatures. Previously published data on PVD nanocomposite coatings have revealed that they can perform better than conventional PVD coatings [20,21].

The second design approach involves a different category of coatings known as adaptive coatings, which have a dual set of characteristics. On the one hand, they are also capable of sustaining heavy loads and temperatures under operation [22]. The surface of an adaptive coating, on the other hand, can easily respond to changing external conditions during friction through the formation of protective/lubricating surface nano-scale tribo-films [23]. These tribo-films alter the frictional and thermal conditions within the cutting zone, reducing the coefficient of friction and thermal conductivity, thereby providing effective protection and lubrication to the operating surface, which leads to a considerable improvement of tool life.

A family of adaptive nano-crystalline TiAlCrN-based coatings [24,25] with (Si+Y) addition has been recently developed for the machining of hard-to-cut materials. These coatings have a hardness of 25–35 GPa and high oxidation stability at elevated temperatures [26]. The effects of silicon (Si) and yttrium (Y) additions in these coatings are different. The addition of Si results in grain refinement [27] and the inclusion of yttrium in a TiACrN coating, which already contains Si prevents intensive grain coarsening at elevated temperatures [28]. Moreover, these elements enhance the beneficial physicochemical reactions at the friction surface, which leads to wear rate reduction [24]. This family of coatings shows promise for dry high-speed machining of hardened tool steels as well as the machining of Ni-based aerospace alloys [22].

Nano-multilayer designs have been recently developed to improve the properties and machining performance of cutting tool coatings [29,30,31]. Such multilayered coatings possess alternating nano-layers with different characteristics [32]. The newly developed nano-multilayered TiAlCrSiYN/TiAlCrN family of coatings consists of alternating 20–40 nm thick nano-layers with a modulating TiAlCrSiYN/TiAlCrN chemical composition and a similar cubic (B 1) crystal structure, in addition to high hardness [22]. The wear performance of this coating layer has been further enhanced through the improvement of its architecture [33]. As a result of this approach, bi-nano-multilayer coatings with a 500 nm thick TiAlCrN sublayer have been introduced [33].

The main design principle behind the alloying of this new bi-nano-multilayer coating is to produce a surface engineered layer with a single-phase (B1) structure and a greater quantity of alloying elements (up to 5% each of Si and Y) for the purpose of generating more of the desired protective/lubricious tribo-films. Studies previously performed on TiAlSiYN/TiAlCrN-based coatings had shown that various thermal barrier tribo-films form on the friction surface. These films consist of mostly sapphire and mullite crystal structures [28,34]. Recently, another type of thermal barrier tribo-film has been identified, which consists of garnet [34]. All of these tribo-films function in synergy to provide protection and lubrication to the tool surface, resulting in a noticeable improvement in its wear performance [34]. 

This paper presents comparative investigations of the structure, properties, and wear performance of these newly developed bi-nano-multilayered TiAlCrSiYN/TiAlCrN coatings during the machining of Ni-based superalloys, such as DA Inconel 718.

## 2. Research Methodology

Three bi-multilayer coatings with varying compositions: Ti_0.2_Al_0.55_Cr_0.2_Si_0.03_Y_0.02_N/Ti_0.25_Al_0.65_Cr_0.1_N (coating A); Ti_0.2_Al_0.52_Cr_0.2_Si_0.08_N/Ti_0.25_Al_0.65_Cr_0.1_N (coating B) and Ti_0.18_Al_0.55_Cr_0.17_Si _0.05_Y_0.05_N/Ti_0.25_Al_0.65_Cr_0.1_N (coating C) were deposited using corresponding targets fabricated by a powdered metallurgical process.

Mirror polished cemented carbide WC-Co substrates (Mitsubishi, UTi20T 120408, Tokyo, Japan) were selected for coating characterization and Kennametal K313 inserts (CNGG432FS) were chosen for cutting tool life studies on a CNC lathe. Coatings were deposited in an R&D-type hybrid PVD coater (AIP-SS002, Kobe Steel Ltd., Kobe, Japan) using a cathodic arc source. Samples were heated up to about 500 °C and cleaned through an Ar ion etching process. N2 mixture gas was fed to the chamber at a pressure of 4 Pa. The arc source was operated at 150 A on a target with a diameter of 100 mm and a thickness of 16 mm. The bias voltage was set to 150 V and substrate rotation to 5 rpm. The thickness of the coating was measured by the ball cratering method. The average thickness of the studied coatings was around 2 μm in the film characterization and cutting tests.

The micro-mechanical characteristics of the coatings were measured on WC-Co coupons using a Micro Materials NanoTest system. Nano-indentation was performed in a load-controlled mode with a Berkovich diamond indenter calibrated for load, displacement, frame compliance, and indenter shape according to the ISO 14577-4 procedure. The area function used by the indenter was determined by indentations of 0.5–500 mN on a fused silica reference sample. The peak load used for coating nano-indentations was 40 mN. A total of 40 indentations were performed on each coating. This load was selected to minimize the influence of any surface roughness on the data while ensuring that the indentation contact depth has remained under 1/10 of the film thickness so that only the coating (load-invariant) hardness would be measured in combination with coating-dominated elastic modulus. Nano-indentation was performed at room temperature. Scratch tests were conducted with an Anton Paar-RST3 Revetest^®^ scratch tester (Graz, Austria). A Rockwell diamond indenter with a 20 μm end radius was used for the tests, which were performed in the progressive mode. A three-scan procedure was adopted for all tests. This procedure consisted of a pre-topography scan at a 0.5 N load, a progressive load scratch scan during which the load was steadily increased from 0.5 N to 5 N, and a post-topography scan at a 0.5 N load. The total scratch length was 0.5 mm with a ramping load rate of 7.02 N/min and a scan speed of 0.78 mm/min. Three scratch tests were performed for each sample. 

The cutting tool life was studied under conditions of finish turning. Coated Kennametal K313 cemented carbide inserts were used for the tool life tests. The cutting experiments were performed on a Nakamura SC450 turning center. Cutting data are presented in Table 1. The cutting tests were performed on advanced workpiece materials such as direct aged Inconel 718 alloy (Table 2). For the machining studies, 125 mm × 800 mm DA 718 disks were used.

Cutting conditions were optimized in our previous research [34] to achieve better tool life and improved productivity. At least three cutting tests were performed for each type of coating under corresponding operations. The error margin of the tool life measurements was approximately 5%.

To evaluate chip characteristics, the chips were collected at the beginning of the cutting (approximately 50 m of cutting length).

The structural and phase transformation at the cutting tool/workpiece interface, as well as the chemical nature of the generated tribo-films, was investigated by a Physical Electronics (PHI) Quantera II (Physical Electronics Inc., Chanhassen, MN, USA) X-ray photoelectron spectroscope equipped with a hemispherical energy analyzer, an Al anode source for X-ray generation, and a quartz crystal monochromator for focusing the generated X-rays. A monochromatic Al K X-ray (1486.7 eV) source was operated at 50 W–15 kV. The system base pressure was as low as 1.0_0_9 Torr, with an operating pressure that did not exceed 2.0_10_8 Torr. The samples had been sputter-cleaned for four minutes by a 4 kV Ar+ beam before any spectra were collected from them. A pass energy of 280 eV was used to obtain all survey spectra and a pass energy of 69 eV was used to collect all high-resolution data. All spectra were obtained at a 45° take-off angle using a dual-beam charge compensation system to ensure the neutralization of all samples. The instrument was calibrated with a freshly cleaned Ag reference foil, with the Ag 3d5/2 peak being set at 368 eV. All data analyses were performed on PHI Multipak version 9.4.0.7 software (Physical Electronics Inc., Chanhassen, MN, USA). Regions undergoing high-resolution (HR) analysis were selected based on careful preliminary investigation of the general photoelectron spectra of the worn surface close to the buildup edge.

## 3. Results and Discussion

The structure of the coatings was studied by the XRD method. The X-ray diffractograms of each coating are presented in Figure 1. 

Black circle labels on the peaks signify the contribution from the carbide substrate. Although a precise XRD analysis was performed, WC-Co substrate signals could also be detected in the studied multilayer coatings with a thickness of 2 μm. Apart from the substrate contribution, the as-deposited coatings feature diffraction peaks that correspond to an fcc NaCl-type structure [35]. Individual intense peaks for fcc CrN (JCPCD 76-2494), Cr_2_N (JCPCD 35-0803), hep-AIN (JCPCD 25-1133), and fcc AIN listed in the database (JCPCD 25-1495) [36] were not found. The TiN, AlN, and CrN reflections became superimposed, generating wide lines near 37°, 63°, and 43°, respectively, signifying the formation of a solid solution [37]. Most of the reflections indicated a single-phase B1 FCC lattice structure in TiAlCrSiYN/TiAlCrN. The study of line intensities (111) and (200) is of particular interest. The intensity of the (111) line is greater than that of the (200) line, which differs from most of the results obtained during XRD investigations of TiAlCrN-based coatings [38,39]. One of the factors contributing to the high intensity of the C (111) line compared to that of the C (200) line may be the texture of the coatings. In highly alloyed and heavily stressed coatings, the strain energy can lead to texture growth. The lowest strain energy becomes the main driving force behind the preferred growth orientation [40]. In this case, the lattice plane with minimal elastic modulus will be oriented in parallel with the substrate. In Bl FCC structures, this corresponds to the (111) plane. It is very likely that the formation of TiAlCrSiYN/TiAlCrN coatings is governed by the strain energy. Another factor determining the high intensity of the (111) line is the presence of atoms with a high X-ray scattering factor in the cubic plane (111). It was found earlier that if the titanium content increases from 17 to 46 at.% and the aluminum content decreases from 53 at.% to 20 at.% in TiAlCrN-based coatings, the intensity of the (200) line grows significantly in comparison with the (111) line. The inclusion of up to 3 at.% Si reduces the intensity of the (200) line by an order of magnitude [41,42]. In nonstoichiometric (TiAlCrSiY) nitride coatings, the total amounts of Si + Y additives were 5.0, 8.0, and 10.0 at.%, which resulted in an increase in the C (111) diffraction line’s intensity compared with that of the (200) line by 1.3, 1.4, and 1.5 times, respectively. Atomic scattering coefficients of Y, Si, and Ti correspond to the values of 19.0, 6.1, and 9.5 under the selected XRD analysis conditions [43]. When the complex nitride was modified it is likely that the Si and Y atoms replaced the Ti atoms in the (111) planes.

The results presented show that coating C, which contains five alloying elements in amounts equal to or above 5 at.%, has a mostly FCC B1 structure and could therefore be classified as a high-entropy alloyed coating. As a result, this coating is considered to be an adaptive high-entropy alloyed coating [44]. 

The following micro-mechanical characteristics were measured: micro-hardness, elastic modulus, H/E ratio; H^3^/E^2^ ratio, and adhesion to the substrate (Table 3). 

The data presented in Table 2 show that the adhesion of each studied coating to the carbide substrate is fairly similar. Coating C has high enough hardness, loading support and high-temperature strength to effectively sustain the machining conditions of Inconel DA 718 [6]. 

Figure 2 presents flank wear vs. length of cut data. The newly developed Ti_0.18_Al_0.55_Cr_0.17_Si_0.05_Y_0.05_N/Ti_0.25_Al_0.65_Cr_0.1_N bi-nano-multilayer coating (coating C) has a 2.08 times longer tool life than the industrial benchmark Al_0.6_Ti_0.4_N (KC 5010) and the tool life of the Ti_0.2_Al_0.52_Cr_0.2_Si_0.08_N/Ti_0.25_Al_0.65_Cr_0.1_N bi-nano-multilayer coating with increased Si content (coating B) and is 40% longer than that of the Ti_0.2_Al_0.55_Cr_0.2_Si_0.03_Y_0.02_N/Ti_0.25_Al_0.65_Cr_0.1_N bi-nano-multilayer coating (coating A). Optical microscopy images show a clear difference in the wear patterns. Intensive crater wear is typical in all of the studied coatings with the exception of the newly developed coating C. The intensity of buildup edge formation is also diminished in this coating. 

It is worth noting that a sharp increase in the wear intensity occurs at a specific cutting length for each coating, as can be seen in Figure 2 (defined as the tangent of the slope of the tangent curve and the horizontal axis). This takes place at a length of cut of 515 m in the industrial benchmark coating (KC 5010), 884 m in the nano-multilayer (B) coating, and 1998 m in the nano-multilayer previous state-of-the-art (A) coating. The wear rate increases by nearly 5 times in the KC 5010 and nano-multilayer (B) coatings and by about 11 times in coating (A). The new (C) nano-multilayer coating behaves differently. At a cutting length of 1180 m, the wear rate increases by 2.3 times, decreases by 2.3 times at a cutting length of 1649 m, increases again by 7.7 times at a cutting length of 2114 m, decreases by 3.5 times at a cutting length of 2346 m, increases by 6.2 times at a cutting length of 2692 m and finally decreases by 5.2 times at a cutting length of 2808 m. A sharp decrease in the wear rate, of around 3.3 times, occurs for the first time at a cutting length of 125 m in each coating. Given that the self-organization process is characterized by a dramatic drop in the wear rate due to the beneficial tribo-film formation [34], it is reasonable to assume that at a cutting length of 125 m, self-organization takes place in all of the coatings. The shapes of the curves in Figure 2 indicate no major fluctuations of the wear rate in coatings KC 5010, nano-multilayer (B) and state-of-the-art (A) during the entire test period until a rapid wear rate increase at the end of the tests. This implies that dissipative structures produced by self-organization (tribo-films; see below) have become depleted at a cutting length of 125 m and that no new ones are being formed. According to the shape of the curve in Figure 2, the self-organization process occurs four times in the new state-of-the-art nano-multilayer (C) coating, characterized by either the formation of new dissipative structures or the intensification of existing ones. This enables the new state-of-the-art nano-multilayer (C) coating to retain a reasonably low wear rate for a substantially longer time than that of the previous coatings. In contrast to the other coatings tested, the new nano-multilayer (C) coating is capable of generating beneficial tribo-films multiple times throughout the entire cutting process. This can be attributed to the enhanced adaptive response [45] of the coating C layer to the buildup edge formation during the machining of Inconel DA 718 [46]. Improved adaptability at elevated temperatures is a typical feature of high-entropy coatings [47]. 

The tribological behavior of the coatings was assessed through a study of chip characteristics (Table 4; Figure 3).

The data are presented in Table 4. 

The chip compression ratio in the C coating is higher, which results in a greater shear plane angle [6]. The shearing force acting on the C coating chips is therefore reduced, which in turn decreases the cutting forces at the tool/chip interface. Chip sliding velocity is higher in the C chips, indicating a shorter tool/chip contact length and a smaller corresponding temperature increase in the cutting zone. Lower friction [48,49] at the tool/chip interface is also confirmed by the higher chip sliding velocity. As a consequence, the coefficient of friction at the tool surface of coating C is reduced. Chip characteristics show less intensive friction at the tool/chip interface, resulting in a longer tool life of the C coating. All of this indicates an overall improvement of the chip flow process in the C coating. 

Figure 3 displays the following chip characteristics: (a) chip type, (b) undersurface morphology, and (c) shear band lines on the upper surface of the chips formed by the coated tools. The chips produced by the tool with the new coating C have superior curling (Figure 3a–d) and a smoother undersurface morphology compared to the previous state-of-the-art coating A (Figure 3b–e). Additionally, the segments of each compared chip are curved in a slightly different manner (Figure 3c–f). The chips produced by tools with coatings A and C have a regular lamellar structure. However, chips produced by the new coating C (Figure 3f) indicate the presence of milder friction conditions at the tool/chip interface. Conversely, discontinuous and brittle shear bands were observed in the chips produced by the tool with the previous coating A (Figure 3c). This is less prominent in the tool with the new coating C (Figure 3f). 

SEM images of the chips’ cross-sections presented in Figure 4 also indicate a significant difference in the tribological behavior of the studied coatings. Chips produced by the tool with coating C have a significantly thinner and curlier shape (Figure 4a,b), which is directly related to the tribological performance of the coated tool. The low thermal conductivity of Inconel DA 718 alloy leads to a very strong heat concentration at the tool/chip interface, causing intensive metal flow in this area [47]. This process is more intense in the chips formed by the tool with the new coating C. It can be concluded from the chip’s characteristics that the surface of the tool with the new coating C is better protected (see details below). This results in a beneficial heat redistribution because more heat is redirected into the chip, which softens the chip/tool interface [50]. The load on the surface is decreased, but the temperature rises at the interface (Figure 4). Moreover, due to the high temperature at the interface, initial stages of transformation from NbC to Nb-rich Laves phases, typical for high-speed Inconel machining [50], most likely occur within the layer of this chip [50,51]. Intensive deformation twinning was observed in the Nb carbides within the chips generated by the tool with coating A (Figure 4a). It is known that the Laves phases are very brittle [52,53]. Therefore, this phase formation (even during the initial stage) promotes crack formation and leads to the intensive fracture of the hard Nb-based phase during the cutting process (Figure 4b) [41]. This accounts for the lower amount of carbides (Figure 4b) at the chip/tool interface, where the temperature is particularly high. Therefore, metal flow within the chip was strongly enhanced in the tool with coating C. 

As was outlined above, this difference in tribological performance is directly associated with the formation of tribo-films on the friction surface. XPS data of the worn surface show that several types of tribo-films are generated during friction (Figure 5): (1) thermal barriers: Al_2_O_3_ sapphire, Al_6_ Si_2_O_13_ mullite, and Y_3_Al_5_O_12_ garnet; (2) thermal barrier/lubricating: Y_2_O_3_, SiO_2_. 

The amount of tribo-films produced is different in the previous state-of-the-art and the newly developed coating. The overall quantity of protective and lubricating tribo-films is noticeably higher in the new coating C than in coating A (Figure 6). This results in superior surface protection/lubrication and overall improvement in the tribological properties of the coatings (for the chip’s characteristics, see Table 3).

However, the overall process was found to be even more complex. In both of the chips studied, softening occurred at the tool/chip interface, whereas hardening took place away from it (Figure 7).

This beneficial behavior of the chips directly corresponds to the formation of thermal barrier tribo-films on the cutting tool rake surface (see Figure 5 and Figure 6). The thermal barrier layer of the tribo-films decreases the heat flow from the cutting zone toward the tool surface [26]. A greater amount of heat is thus transferred into the chips, softening the surface area (Figure 7). On the other hand, the chips produced by coating C are accompanied by a very intense metal flow (due to a lower coefficient of friction during cutting; Table 3), resulting in the formation of strain-induced martensite zones (Figure 4b) within the surface area of the chips [29], which is a self-organization process [54]. One of the major features of Inconel DA 718 machining is its tendency to work harden at high operating temperatures [55,56]. If the chips produced by the tool with coating C are compared with those produced by the tool with coating A, it can be found that the increased hardness of the surface layer of the chips produced by the tool with coating C reduces the adhesion of the workpiece material to the tool surface. This, in turn, results in the formation of curlier chips (Figure 4b), which directly corresponds to superior wear performance. 

## 4. Conclusions

A comprehensive wear performance study was carried out on several TiAlCrSiN/TiAlCrN-based bi-nano-multilayer coatings with different chemical compositions during the machining of DA 718 alloy. 

The following conclusions have been drawn based on the obtained results: The newly developed wear-resistant bi-nano-multilayer Ti_0.18_Al_0.55_Cr_0.17_Si_0.05_Y_0.05_N/Ti_0.25_Al_0.65_Cr_0.1_N coating (C) contains five alloying elements in amounts equal to or above 5 at.% and has a mostly FCC B1 structure. As such, it can be classified as an adaptive high-entropy alloyed coating.The most crucial feature of coating C is its enhanced adaptive performance during the machining of Inconel DA 718 alloy, as evaluated by its ability to form a specific set of tribo-films during operation. The quantity of beneficial tribo-films which form on the friction surface of the tool is greater in this coating compared with the previous state-of-the-art coating A. The formation of such tribo-films enhances the protection and lubrication of the cutting tool’s surfaces, thereby reducing the intensity of wear.The more complex features of the tribological process were evaluated through tribological studies. Increased metal flow and lower chip hardness within the tool/chip contact zone were a consequence of the improved surface protection and lubrication provided by the formation of tribo-films on the friction surface. A simultaneous phase transformation takes place within the layer of the chip due to intense metal flow (due to the formation of strain-induced martensite zones during severe plastic deformation), causing the chip material to harden. This in turn reduces workpiece material adhesion to the tool surface, thereby improving the wear performance of the newly developed adaptive coating C.A new nanomaterial research strategy is proposed with the goal of enhancing the various multi-scale self-organization processes taking place during cutting. These self-organization processes consist of the nano-scale tribo-film formation on the friction surface in conjunction with the micro-scale generation of strain-induced martensite zones within the layer of the chips. (An optimized combination of these processes can considerably enhance the wear performance of the newly developed PVD bi-nano-multilayer Ti_0.18_Al_0.55_Cr_0.17_Si _0.05_Y_0.05_N/Ti_0.25_Al_0.65_Cr_0.1_N high-entropy adaptive coating (coating C).

## Figures and Tables

**Figure 1 materials-15-01329-f001:**
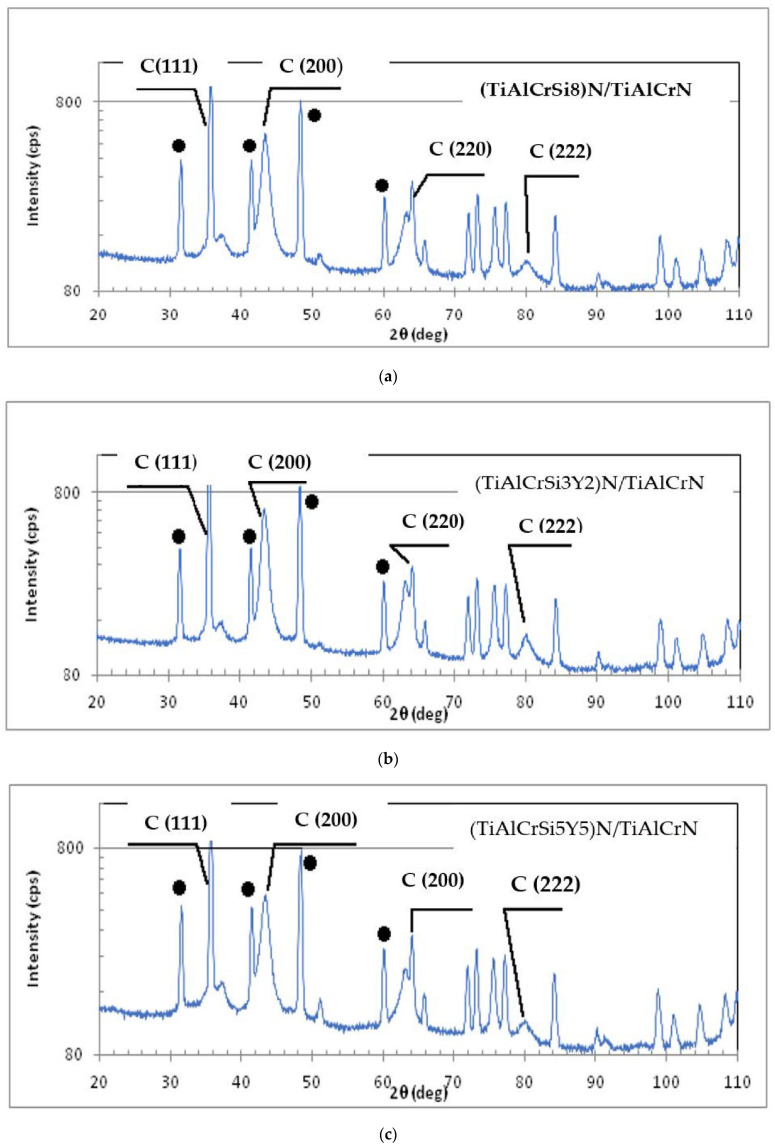
X-ray diffractograms of the studied PVD coatings: (**a**) bi-multilayer TiAlCrN/TiCrAl52Si8N coating (coating B); (**b**) bi-multilayer TiAlCrN/TiCrAlSi3Y2N coating (coating A); (**c**) bi- multilayer TiAlCrN/TiCrAlSi5Y5N coating (coating C).

**Figure 2 materials-15-01329-f002:**
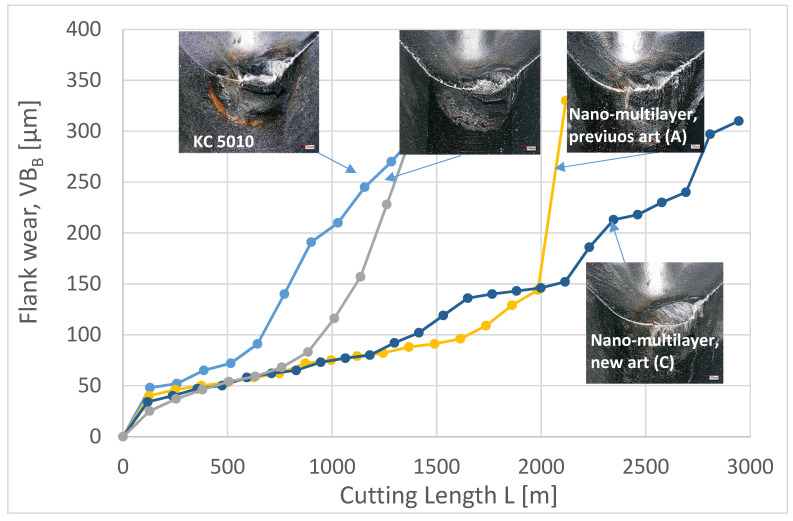
Flank wear vs. length of cut data of the studied coatings with the optical microscopy images of the worn surface.

**Figure 3 materials-15-01329-f003:**
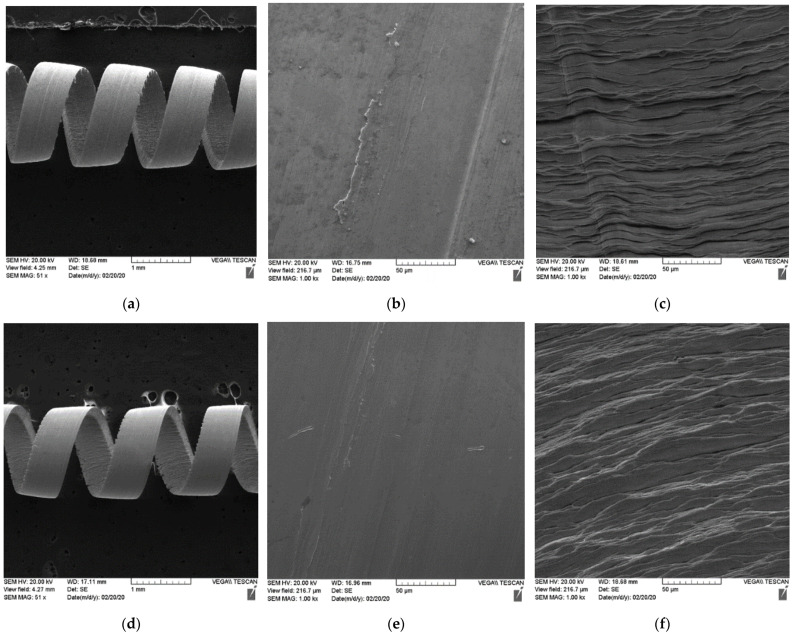
Chip characteristics for the tools coated by multilayer TiAlCrN/TiCrAlSi3Y2N multilayer coating (previous art): (**a**–**c**) new multilayer TiAlCrN/TiCrAlSi5Y5N coating: (**d**–**f**); (**a**,**d**) general view: chips curling; (**b**,**e**) chip undersurface morphology; (**c**,**f**) shear band lines of the previous state-of-the-art and the new coating.

**Figure 4 materials-15-01329-f004:**
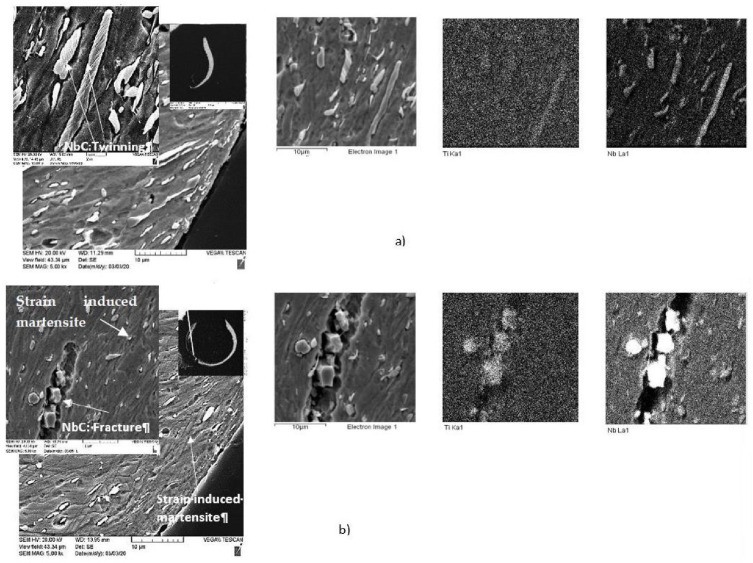
SEM/EDS data of the chip cross-sections: (**a**) the TiAlCrN/TiCrAlSi3Y2N bi-multilayer coating (previous state-of-the-art); (**b**) the new bi-multilayer TiAlCrN/TiCrAlSi5Y5N coating.

**Figure 5 materials-15-01329-f005:**
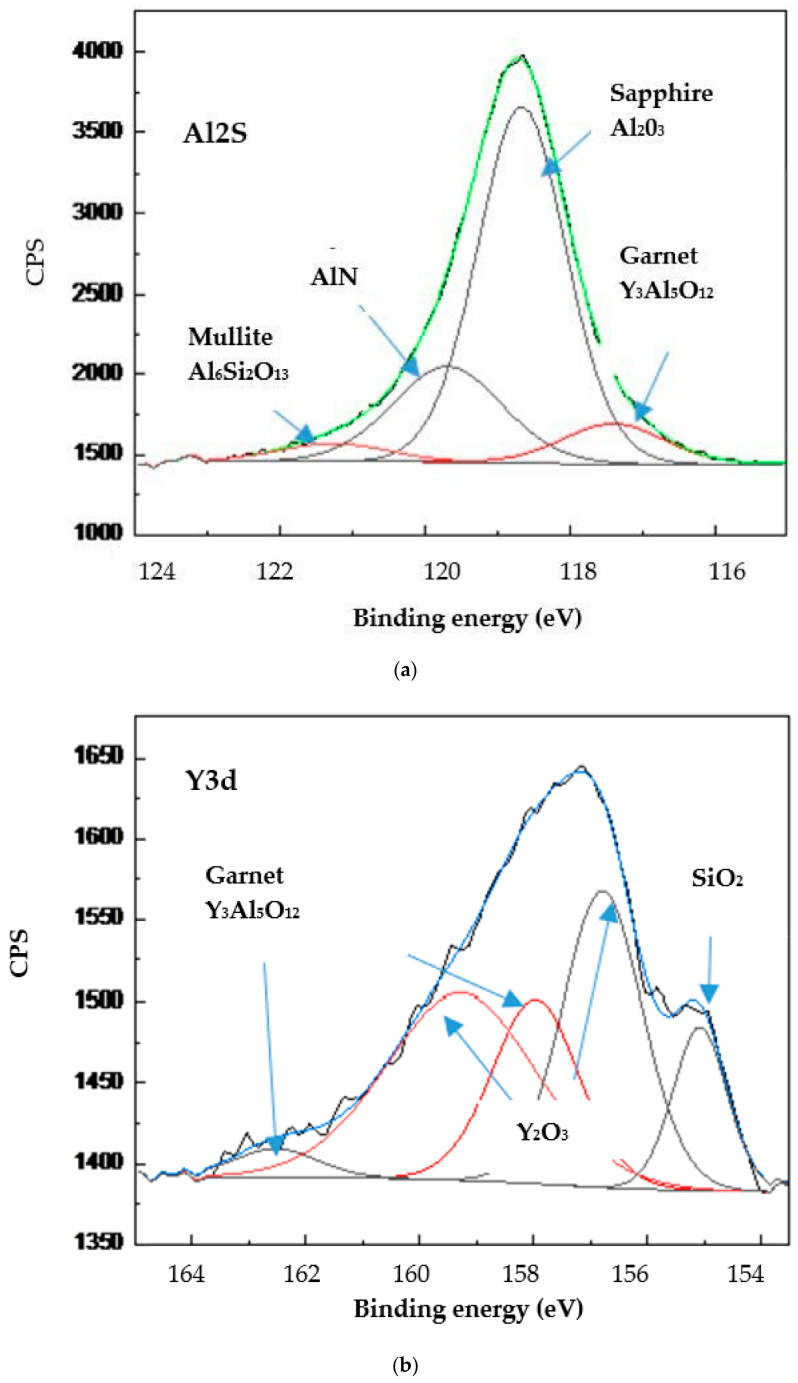
Typical photoelectron spectra of the tribo-oxides in the wear zone of the studied coatings: (**a**) Al-2s spectrum; (**b**) Y-3d spectrum.

**Figure 6 materials-15-01329-f006:**
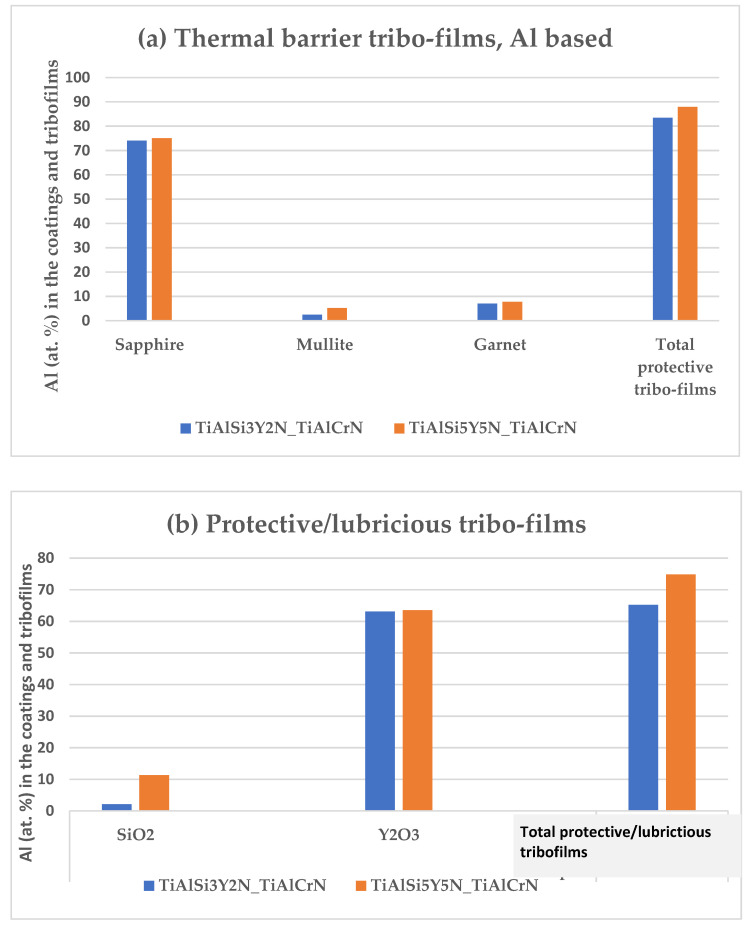
Comparative amounts of tribo-films in each coating. XPS data.

**Figure 7 materials-15-01329-f007:**
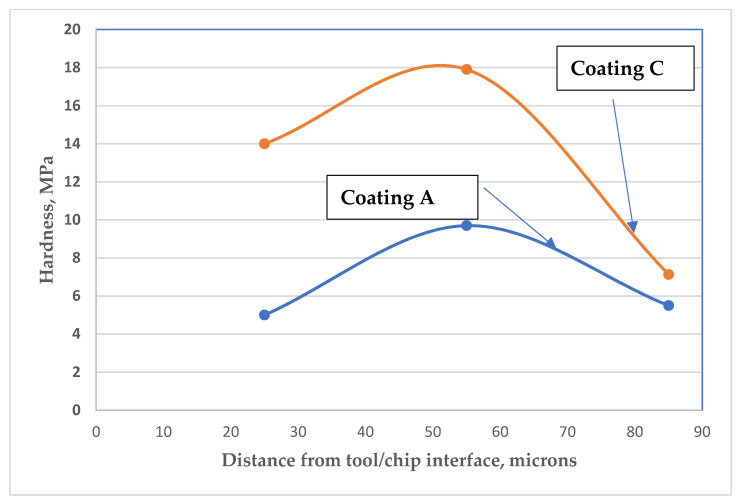
Chip hardness distribution for tools with a TiAlCrN/TiCrAlSi3Y2N bi-multilayer coating (previous state-of-the-art) and new bi-multilayer TiAlCrN/TiCrAlSi5Y5N coating.

**Table 1 materials-15-01329-t001:** Cutting data for the experiments performed.

Cutting Data						
Machining Operation	Cutting Tool Substrates	Workpiece Material	Hardness	Speed, m/min	Feed, mm/rev	Depth of Cut, mm
Turning	Kennametal K313 carbide turning inserts	Direct aged Inconel 718	HRC 47–48	60	0.125	0.25

**Table 2 materials-15-01329-t002:** Inconel DA 718 composition.

Major Elements Alloy Content, Weight %			
Cr	Ni	Nb	Ti
18.5	52.9	3.05	089

**Table 3 materials-15-01329-t003:** Micro-mechanical characteristics of the studied coatings at room temperature.

Coatings	Hardness, GPa	Reduced Elastic Modulus, GPa	H/E_r_ Ratio	H^3^/E_r_^2^ Ratio	Adhesion to the Substrate, Lc_2_, N
Ti_0.2_Al_0.55_Cr_0.2_Si_0.03_Y_0.02_N/Ti_0.25_Al_0.65_Cr_0.1_N Multilayer (coating A)	28.4 ± 4.8	361.1 ± 36	0.0789	0.1774	55.22
Ti_0.2_ Al _0.52_Cr_0.2_Si_0.08_N/Ti_0.25_Al_0.65_Cr_0.1_N Multilayer (coating B)	39.7 ± 4.5	427.2 + 36	0.0929	0.3431	50.19
Ti_0.18_Al_0.55_Cr_0.17_Si_0.05_Y_0.05_N/Ti_0.25_Al_0.65_Cr_0.1_N Multilayer (coating C)	37.6 ± 4.3	417.3 + 34	0.0904	0.3087	55.09

**Table 4 materials-15-01329-t004:** Chip characteristics of the studied coated tools.

Coating	Tribological Characteristics				
	Chip Compression Ratio	Share Angle (°)	Share Strain	Chips Sliding Velocity	Coefficient of Friction
Coating A	1.16	38.62	2.204	69.35	0.33
Coating C	1.30	40.29	2.189	78.36	0.17

## Data Availability

Not applicable.

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
