# Peer review of "Enhancement of Multi-Scale Self-Organization Processes during Inconel DA 718 Machining through the Optimization of TiAlCrSiN/TiAlCrN Bi-Nano-Multilayer Coating Characteristics"

_materials, 2022, doi:10.3390/ma15041329_

Round 1

Reviewer 1 Report

This paper presents a comparative investigations of the structure, properties, and wear performance of bi-nano-multilayered TiAlCrSiYN/TiAlCrN coatings during machining of Ni-based superalloys, such as DA Inconel 718.

Although the work is very interesting, has a lot of information and is well structured, it requires some improvements:

  • It is not correct to make appointments of the form [1–3]. Each reference must be commented on individually.
  • If the insert manufacturers preferentially apply the CVD coating process. Why is it stated that “Surface engineering of cemented carbide cutting tools can be an effective solution to the problems posed by the machining of these hard-to-cut materials, particularly through the application of Physical Vapor Deposited (PVD) coatings”?
  • Explain the chemical composition and diameter of the machined material.
  • Why did you use a Mitsubishi insert as sustrate for the characterization of the coatings and a Kennametal K313 insert as sustrate to coat them and use in machining tests?
  • How is it that figure 2 speaks of a quality of Kennametal KC 5010?
  • Since the depth of cut is so small (0.25 mm) that it produces a small worn area. That flank wear usually has a very irregular shape, how is it possible to ensure the sayings of lines 228 to 239?
  • Explain how the values ​​indicated in table 3 were obtained and its meaning. Why only the values ​​corresponding to coatings A and C appear?
  • Figures 5, 6 and 7 are very ugly.

Author Response

  • It is not correct to make appointments of the form [1–3]. Each reference must be commented on individually.

Response. Correction is made

  • If the insert manufacturers preferentially apply the CVD coating process. Why is it stated that “Surface engineering of cemented carbide cutting tools can be an effective solution to the problems posed by the machining of these hard-to-cut materials, particularly through the application of Physical Vapor Deposited (PVD) coatings”?

Response. Both coatings deposition methods are widely used for cutting tool production.

  • Explain the chemical composition and diameter of the machined material.

Response. Correction is made. Table 2 is added.

  • Why did you use a Mitsubishi insert as sustrate for the characterization of the coatings and a Kennametal K313 insert as sustrate to coat them and use in machining tests?

Response. These two substrates have identical chemical compositions and structure.

  • How is it that figure 2 speaks of a quality of Kennametal KC 5010?

Response. Figure 2 shows wear intensity and wear patterns. For the studied severe conditions, the quality of KC 5010 is normal.

  • Since the depth of cut is so small (0.25 mm) that it produces a small worn area. That flank wear usually has a very irregular shape, how is it possible to ensure the sayings of lines 228 to 239?

Response. Flank wear measured was maximum one. As it is outlined in the text all coated tool had three stages of flank wear: initial, stable, catastrophic. However, coating C behaves differently. As it is shown in the text: This can be attributed to the enhanced adaptive response [ 53] of the coating C layer to  buildup edge formation during the machining of Inconel DA 718 [54].

  • Explain how the values ​​indicated in table 3 were obtained and its meaning. Why only the values ​​corresponding to coatings A and C appear?

Response. Citation is added with the methodological source of the tribological properties characterization using chips characteristics studies. Only coatings A and C has been studied because coating B shows low tool life, similar to industrial benchmark.

  • Figures 5, 6 and 7 are very ugly.

Corrections are made in corresponding figures according to reviewer comments. Presentation is improved as much as possible.

Reviewer 2 Report

The work deals with very important issues of nano-layers of materials based on titanium. The work is very innovative.

There is no explanation for filling the scientific gap and the scientific value of the work between the final lines.

The summary can be shortened slightly. The authors described what they did but did not indicate how innovative it is in relation to the other works.

The literature review is very modest and a lot is missing in this section. It lists the most important information in this field and works. However, there is no broader view of the whole in relation to the topic of electromobility.

Perhaps it is worth mentioning a few works in this field. https://doi.org/10.3390/ma14226839; doi: 10.4271 / 03-14-06-0051;

The second part deals with the continuation and description of the achievement. This is not the methodical part. Part 2 and 3 are prepared correctly. Very brief and to the point. No unnecessary information. Therefore, I believe that this part should not be corrected. The drawings of the patterns are correct. I have attention. The summary is ok.

Bugs in work: Line 107 - shouldn't there be a research methodology in line with the MDPI chapter layout. Adapt font sizes and format to MDPI style. Figure 1 - some descriptions not too clear in the drawings. Varying the font size. Table 2. Wrong style, incompatible with MDPI. Figure 2. Too Big. Figure 5. Wrong size too big. Adjust the style of Figures 6 and 7 to the rest of the drawings. Invalid MDPI font style.      

Author Response

There is no explanation for filling the scientific gap and the scientific value of the work between the final lines.

Response. All these issues are outlined in the conclusion.

The summary can be shortened slightly. The authors described what they did but did not indicate how innovative it is in relation to the other works.

Response. All these issues are outlined in the conclusion,

The literature review is very modest and a lot is missing in this section. It lists the most important information in this field and works. However, there is no broader view of the whole in relation to the topic of electromobility.

Perhaps it is worth mentioning a few works in this field. https://doi.org/10.3390/ma14226839; doi: 10.4271 / 03-14-06-0051;

Response. Correction is made according to reviewer comment. Corrsponding works are included in the Reference section

The second part deals with the continuation and description of the achievement. This is not the methodical part. Part 2 and 3 are prepared correctly. Very brief and to the point. No unnecessary information. Therefore, I believe that this part should not be corrected. The drawings of the patterns are correct. I have attention. The summary is ok.

Bugs in work: Line 107 - shouldn't there be a research methodology in line with the MDPI chapter layout.

Adapt font sizes and format to MDPI style.

Response: correction is made.

Figure 1 - some descriptions not too clear in the drawings. Varying the font size.

Response. We believe we presented the data in the best clear way.

Table 2. Wrong style, incompatible with MDPI.

Response. We presented similar data in many journals and no questions were asked. We do not understand what is wrong.

Figure 2. Too Big.

Response. We do not understand the question. The figure does not look big to us.

Figure 5. Wrong size too big.

Response. We do not understand the question. The figure does not look big to us. Correction is made. Size of HR spectrum names is reduced.

Adjust the style of Figures 6 and 7 to the rest of the drawings. Invalid MDPI font style.  

Correction is made. Font is changed correspondingly.   

Round 2

Reviewer 1 Report

The work needs to improve three aspects:
1) The introduction: it is very basic since the paper has 63 references.
2) The number of self-citations is incredible (32%)
3) Images 1, 5 and 6 are very ugly.

Author Response

The work needs to improve three aspects:
1) The introduction: it is very basic since the paper has 63 references.

Response to reviewer. The number of references is reduced due to the reduction of self-citations.

We tried to provide a focused literature overview to justify the goal of the research.
2) The number of self-citations is incredible (32%)

Correction is made according to reviewer comment (see above). Number of self-citations is reduced.

3) Images 1, 5 and 6 are very ugly.

Corrections are made according to reviewer comments and MDPI format. We corrected the figures in the best possible way.

Figure 1. We made all possible corrections according to MDPI format.

Figure 5. We made all possible corrections according to MDPI format.

Figure 6. We made all possible corrections according to MDPI format.